# Experimental Study on the Lubrication Enhancement of Slider-on-Disc Contact by Stearic Acid Adsorption under Limited Lubricant Supply

**Yusheng Jian** [1], **Zhaogang Jing** [1], **Feng Guo** [1,*], **Pat Lam Wong** [2] **and Xinming Li** [1]

1   School of Mechanical and Automotive Engineering, Qingdao University of Technology,
    Qingdao 266520, China
2   Department of Mechanical Engineering, City University of Hong Kong, Kowloon, Hong Kong, China
*   Correspondence: mefguo@qut.edu.cn

**Abstract:** The optimization of the lubricant supply quantity contributes to minimizing energy losses and wastage. To enhance the performance of hydrodynamic bearings running with limited lubricant supply (LLS), this study examined the effect of stearic acid as an additive. Stearic acid is commonly used for boundary lubrication as an organic friction modifier. How the stearic acid adsorption affects the hydrodynamic lubrication of a slider-on-disc contact under LLS was investigated using interferometry measurement and fluorescence observation in this study. Firstly, the oil reservoir of PAO10 with stearic acid adsorption was observed at the slider entrance. Secondly, the film thickness versus speed of PAO10 and PAO10 with 0.1% and 0.3% stearic acid, respectively, were measured. Finally, the morphology and surface properties of the glass and steel blocks with stearic acid adsorption were characterized by atomic force microscopy and Fourier transform infrared spectroscopy. The results show that the stearic acid adsorption layer weakens the wettability of the lubrication track and induces the 'dewetting' phenomenon of the lubricating oil. Thus, discrete oil distribution in the form of stripe or droplet can be generated, which leads to the accumulation of lubricating oil at the slider entrance to form a reservoir. An additional inlet pressure that is generated by the oil reservoir due to surface tension increases the oil film formation capacity. Furthermore, the morphologies of the adsorbed stearic acid layer on the glass and the steel blocks are, respectively, characterized by the nano-sized granular bulge and brush structure. This study reveals a new role of stearic acid adsorption in promoting LLS lubrication.

**Keywords:** stearic acid; limited lubricant supply; oil reservoir; film thickness; slider-on-disc contact; hydrodynamic lubrication

## 1. Introduction

According to Holmberg and Erdemir [1], 23% of the world's total energy consumption originates from tribological contacts. With the exception that some of the consumption is for certain purposes, such as traction, the majority is wastage or loss via friction and wear. Oil film lubrication is widely used to reduce such losses. However, common practice tends to apply far more than the necessary quantity of lubricants, due to concerns of lubrication failure, which can be serious and may even lead to the collapse of the whole operating system or machine. That extra provision of lubricating oil is not only a waste but also generates excessive viscous friction owing to the oil flow in the vicinity of the lubricated contact; for example, as in the bearings. Considering the typical dimension of lubricated conjunctions, it requires theoretically only a tiny amount of oil to realize effective lubrication; that is, the total separation of the two lubricated rubbing surfaces. The optimization of the lubricant supply quantity contributes to minimizing energy losses. Thus, there has been considerable interest in recent years; in particular, experimental studies on the effects of the limited lubricant supply (LLS) condition.

The thickness of the hydrodynamic lubricating film in a bearing contact is governed by the Reynolds equation with the assumption of unlimited lubricant supply. Its magnitude depends purely on the dynamic conditions (the load and the speed) and the oil viscosity. It is referred to as the fully flooded condition. When the supply lubricant is limited, leading to the corresponding hydrodynamic film thickness being less than the theoretical fully flooded value, this condition is termed oil starvation. In fact, bearings readily run under oil-starved conditions, especially at high-speed running or with grease lubrication. Oil starvation and the condition when it occurs have been investigated on both the experimental [2–7] and the theoretical sides [5,6,8–10]. Studies have also covered the effects of various parameters, including obviously the amount of lubricant supplied. As expected, the film thickness reduces with LLS. Theoretically, if the buildup film thickness $h$ is three times higher than the composite surface roughness $\sigma$ of the lubricated conjunction (i.e., lambda ratio $h/\sigma \geq 3$ [11]), it guarantees a safe bearing running. Both the experimental and theoretical studies prove that even under LLS conditions, it may still be possible to separate the rubbing surfaces with a thin lubricating oil layer. However, how to sustain it is another issue.

To maximize the limited amount of lubricant supply can be explored from two different aspects: firstly, to increase the actual oil entrainment to the bearing contact from the LLS volume, and, secondly, to enhance the lubrication effect of a given amount of entrained oil. The former is largely an issue at the out-of-contact lubrication track, and the latter is probably a matter inside the bearing contact.

The oil distribution on the out-of-contact lubrication track readily acquires a structure of a free-surface thin oil layer bordered by oil ridges on the two sides, which is formed when the oil passes through the bearing and some (in fact, most) of the oil is displaced sideways. The replenishment (that is, the free viscous oil flow from the two ridges to the central track) may not be effective due to the oil being too thick or the time interval between two consecutive roll- or slide-overs of the bearing being too short. Ali et al. devised a simple but effective means to scrap the displaced lubricant on the two sides back to the depleted track, resulting in robust replenishment [12,13]. Li et al. proposed an even more elegant way to enhance the replenishment by using the interfacial forces at the oil/solid boundary. They used an oleophobic coating on the two sides of the oleophilic lubrication track (of steel or glass surfaces) to produce a wettability gradient that creates unidirectional flow towards the lubrication track (from the oleophobic to the oleophilic regions) [14]. Liu et al. adopted the same concept but the wettability gradient was created by using lasered surface texturing [15]. All of these experiments demonstrated the increase in the lubricating film thickness, which proves that the replenishment, i.e., the entrainment, are enriched by these methods.

Applying surface texturing and oleophobic coating to the bearing surfaces can change the film-forming or load-carrying capacity under hydrodynamic lubrication. For example, Guo et al. used the direct laser interference patterning technique to produce lateral microgrooves in the inlet region of slider bearings [16]. This partial textured bearing surface enables an increase in the film thickness, i.e., load-carrying capacity of the bearing. Recently, Shen et al. numerically analyzed the performance of a hydrodynamically lubricated sliding bearing that has chevron textures on one of the surfaces [17]. The origination, the size and the distribution of the textures were studied. The proposed textured design yields great improvement on lubrication performance. Kalin and Polajnar experimentally studied the effect of Diamond-like Carbon (DLC) coatings on the friction of lubricated steel/DLC-coated rubbing surfaces [18,19]. The friction was reduced that was attributed to the weak affinity between the oil and DLC at the interface, thus leading to slip. On one hand, boundary slip reduces the overall friction of the lubricated contact. On the other hand, it weakens the oil entrainment into the bearing contact, resulting in a decrease in film thickness. To overcome this dilemma, Sun et al. devised two partial-slip surface patterns for slider bearings: lateral central stripe and inverse triangle [20]. The slip patterns were facilitated by oleophobic coatings (or termed as anti-fingerprint (AF) coatings) and their effects were validated experimentally through slider bearing tests. It was demonstrated that

the bearing load-carrying capacity was enhanced beyond the prediction of the Reynolds equation. Apart from the ideas of surface texturing and oleophobic coating, Li et al. recently addressed the issue of LLS lubrication improvement from another perspective: the form or feature of the free-surface of the supplied oil layer [21]. They compared two different profiles of the entrained oil layer: an ordinary uniform free-surface oil layer and an oil-bead featured layer. The results show that the oil-bead layer provides a better LLS lubrication effect in terms of high film thickness.

The oil-bead layer can be actualized via the oil dewetting phenomenon, which requires weak liquid/solid interaction. A thin oil layer swept on an oleophobic surface is very unstable and readily beads up to form oil droplets on the surface. This paper presents an experimental study about how LLS lubrication can be promoted by oil dewetting. The formation of oil beads is restricted to occur within the lubrication track. If the whole bearing surface is oleophobic, oil beads will be formed sporadically, which hinder the oil spreading, reducing the amount of oil replenishment to the lubrication track. If the lubrication track is only coated with AF coating, the result is a negative scenario: a unidirectional interfacial force driving the oil toward the two sides of the track. If any of the supplied oil is displaced by the bearing to the edges of the oleophobic track (covering the boundary between the oleophobic track and the oleophilic outer region), the oil tends to leak away from the track. This study proposes a novel idea of dynamically modifying the lubrication track to oleophobic by using appropriate additives to the oil. The additive forms in-situ an oleophobic adsorption layer of an optimal width. The adsorbed layer can be restored through self-healing when it is removed during the bearing operation. It is like the additive of friction modifiers for boundary lubrication. The polar groups on the rubbing surfaces spontaneously form a tightly packed directional adsorption film. This study uses stearic acid as the additive and PAO10 as the base oil. The adsorption of stearic acid will change the surface to oleophobic, promoting the oil dewetting process on the lubrication track. According to Guo et al. [22], PAOs acquire large contact angles on an AF coated surface, which is more than double large contact angles on an uncoated oleophilic surface, whereas the contact angle hystereses (CAHs) on the two specimen surfaces are large (about 34°) and approximately the same (less than 4% in difference). In fact, Guo et al. identified the interfacial parameter, CAH, as the key parameter correlated with the boundary slip condition [23]. Thus, PAO was chosen in this study for the expected large dewetting phenomenon on the oleophobic adsorption layer and for the large and stable CAHs on both the oleophobic and oleophilic surfaces.

## 2. Experimental Detail

### 2.1. Experimental Apparatus

The optical slider-bearing test rig used in the present study is schematically depicted in Figure 1. More details about the test rig can be found in [24]. The lubricating oil is dragged into the convergence gap by the rotation of the glass disc to generate the hydrodynamic effect. The oil film thicknesses at the inlet and outlet edges of the slider are, respectively, $h_1$, and $h_0$, which can be accurately determined by interferometry with a coherent laser source of wavelength 640 nm. The inclination angle of the slider $\alpha$ between the steel slider and the glass disc, which can be adjusted using the adjustment bolts, is known from the number of interference fringes formed within the bearing contact area. A constant inclined angle was chosen for all the tests of different speeds and loads. The relationship between the number of interference fringes and the angle of slider is described as

$$\alpha = \frac{\lambda N}{2nB} \tag{1}$$

where:

$\alpha$ is the inclination of the slider (rad);
$\lambda$ is the wavelength of incident light (m);
$N$ is the number of interference fringes;

$n$ is the refractive index of lubricating medium;
$B$ is the slider breadth in the $x$-direction, as shown in Figure 1 (m).

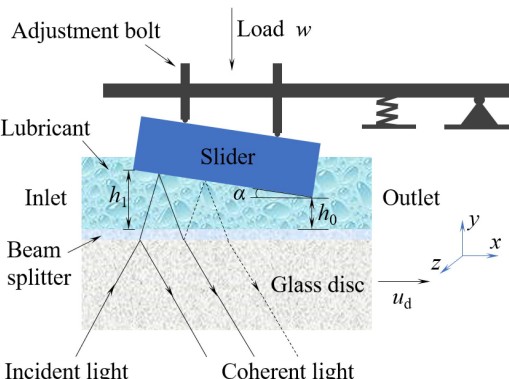

**Figure 1.** Schematic illustration of the optical slider-bearing test rig.

Figure 2 shows the fluorescence system, which consists primarily of an exciter filter, a barrier filter, a dichroic mirror and a fluorescent charge coupled device (CCD) camera (Lumenera Corporation, Inc., Ottawa, ON, Canada) [25]. Light in the wavelength range of 450~500 nm passes through the exciter filter, while fluorescence at wavelengths greater than 500 nm passes through the barrier filter. The lubricant containing the fluorescent agent is illuminated by the blue excitation light to produce fluorescence, which is then imaged with a fluorescent CCD camera. Thus, the lubricant distribution on the free surface of the slider and the lubrication track is mapped. The test bench was shaded to avoid the influence of ambient light source on the test results during the test.

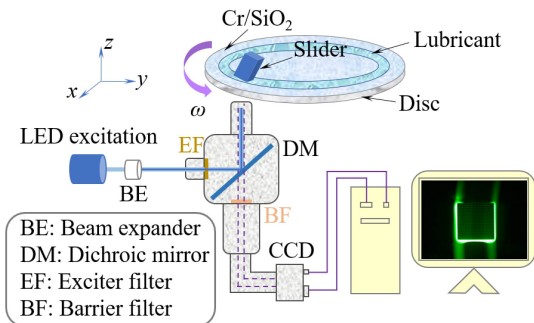

**Figure 2.** Schematic of fluorescence measurement system.

### 2.2. Materials and Experimental Conditions

Sliders are made of bearing grade AISI52100/DIN 100Cr6 steel. The size of sliding surface is 4 mm × 4 mm and its roughness $R_q$ is 0.0114 μm. The glass disc used in the experiment is made of K9 glass, and the working surface is coated with a bi-layer film (Cr layer at the bottom and $SiO_2$ layer on the top) to facilitate interferometry measurements of the lubricating film and provide wear resistance. The diameter of the disc is 140 mm, and its $R_q$ is 0.0042 μm. Furthermore, the working surface of the glass disc was hydroxylated by oxygen plasma for 2 min to promote the stearic acid adsorption on the surface.

Stearic acid (Shanghai Abby Chemical Reagent Co., Ltd., Shanghai, China), whose molecular structure is shown in Figure 3, is the additive to the base oil PAO10. Two other oil samples were prepared with different fractions of stearic acid to the base oil. The fluorescent agent, coumarin-6 with a purity of 98%, was added into the oil samples at a concentration of 0.5 mmol/L. The fluorescent agent was uniformly dissolved in the lubricant by heating and stirring in a magnetic stirrer. The main physical and chemical properties of stearic acid and coumarin-6 are shown in Table 1.

**Figure 3.** Molecular structure of stearic acid.

**Table 1.** Main physical and chemical properties of stearic acid and coumarin-6.

| Material | Molecular Formula | Molecular Weight | Density/(g/cm$^3$) | Melting Point |
|----------|-------------------|------------------|----------------------|---------------|
| Stearic acid | $C_{18}H_{36}O_2$ | 248.48 | 0.847 | 67~70 |
| Coumarin-6 | $C_{20}H_{18}N_2O_2S$ | 350.43 | 1.311 | 205–208 |

Two sets, each of which included three different specimen lubricants, were prepared. Thus, there were six specimen lubricants in total. Each set consisted of PAO10 base oil (PAO10), PAO10 with 0.1 wt% (PAO10S$_{0.1}$) and 0.3 wt% (PAO10S$_{0.3}$) of stearic acid. One set was without any fluorescent agent, while coumarin-6 was added to the other set. The dynamic viscosity and refractive index of these specimen lubricants are shown in Table 2. The addition of small percentages of stearic acid and the fluorescent dye had little effect on the properties of the base oil.

**Table 2.** Dynamic viscosity and refractive index of lubricating oil for testing.

| Lubricant | Dynamic Viscosity $\eta$/(mPas@22 °C) | Refractive Index $n$ |
|-----------|----------------------------------------|----------------------|
| PAO10 | 120.1 | 1.4624 |
| PAO10S$_{0.1}$ | 117.7 | 1.4625 |
| PAO10S$_{0.3}$ | 119.9 | 1.4617 |
| PAO10+coumarin-6 | 116.3 | 1.4625 |
| PAO10S$_{0.1}$+coumarin-6 | 119.2 | 1.4624 |
| PAO10S$_{0.3}$+coumarin-6 | 117.1 | 1.4625 |

Experiments were conducted under the following conditions. The bearing track radius $R_c$ on the glass disc was 42 mm and the sliding speeds $u_d$ were in the range of 1~40 mm/s. Three set loads $w$ of 2 N, 4 N and 6 N were applied. Inclination of the slider was kept constant in the experiments ($\alpha = 8 \times 10^{-4}$ rad). The oil supply volume $q$ was 2 μL. Fully flooded experiments (denoted as 'full') were also carried out for comparison. For LLS conditions, the tiny amount of lubricant was evenly dispensed along the central axis of the lubrication track by micropipette. The lubricant was then uniformly spread on the lubrication track by the slider at a low speed of 0.1 mm/s and a light load of 0.1 N. All the tests were carried out in a controlled lab environment with the temperature of $22 \pm 0.5$ °C and the humidity of $35 \pm 5$% RH after the pre-running spread.

The major challenge of conducting experiments with LLS conditions is the failure of lubrication, leading to damage of the contact surfaces. The lab-based test rig (as schematically shown in Figure 1) resembles a full film hydrodynamic bearing contact of a steel surface and a glass surface. Low loads were thus selected for the tests to facilitate full hydrodynamic films.

## 3. Results and Discussion

### 3.1. Effect of Stearic Acid Adsorption on Morphology and Distribution of Lubricants

To explore the influence of stearic acid adsorption on the morphology and distribution of lubricants under LLS, the following three aspects were observed and analyzed:

1. The morphology of static lubricants on the lubrication track of the glass disc;
2. The morphology of lubricants in the oil reservoir at the slider entrance during the running of the glass disc;
3. The distribution of lubricants at the side edge of the slider.

The static distribution patterns of PAO10, PAO10S$_{0.1}$ and PAO10S$_{0.3}$ with oil supply of 2 μL on the lubrication track were observed. A total of 2 μL of the three specimen

oils was evenly dispensed at the central axis of the track with a micropipette. After the pre-running spread, the slider bearing was run under the conditions of load 2 N and speed 1 mm/s for 20 min. Snap shots of the lubrication track, as shown in Figure 4, were taken afterwards. Pure PAO10 was distributed to cover the whole lubrication track (Figure 4a); the distribution width was slightly larger than the slider width. However, for PAO10 with different mass fractions of stearic acid, dewetting occurred on the track which is probably due to the adsorption of stearic acid on the glass disc surface. As a result, discrete stripes (Figure 4b) or droplets (Figure 4c) were generated. The degree of dewetting was more obvious with the increase in the loading of stearic acid. The discrete stripes or droplets were all inside the track, such that the width of the oil distribution was less than that of the slider.

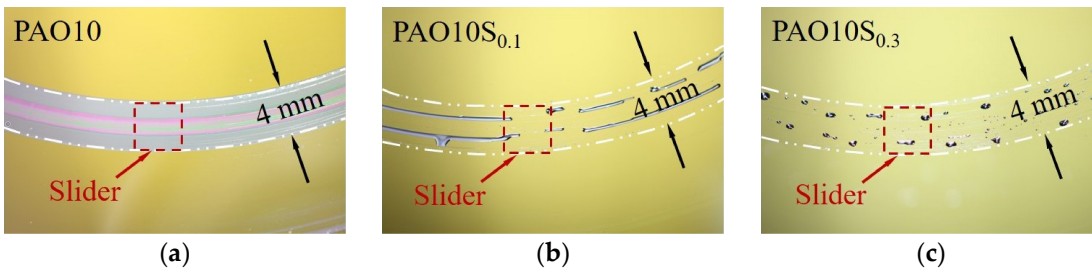

**Figure 4.** Lubricant distribution pattern on glass disc track surface, $w = 2$ N, $q = 2$ µL. (**a**) PAO10, (**b**) PAO10$S_{0.1}$ and (**c**) PAO10$_{0.3}$.

The conditions of oil entrainment at the slider entrance have a very important influence on the lubricating oil film formation. Therefore, the morphology of the oil reservoir of PAO10, PAO10$S_{0.1}$ and PAO10$S_{0.3}$ at the slider entrance was observed and analyzed using the fluorescence method. Figure 5 shows the fluorescence images of the oil reservoir of the three lubricating oils at the slider entrance at various sliding speeds with load of 2 N and oil supply of 2 µL.

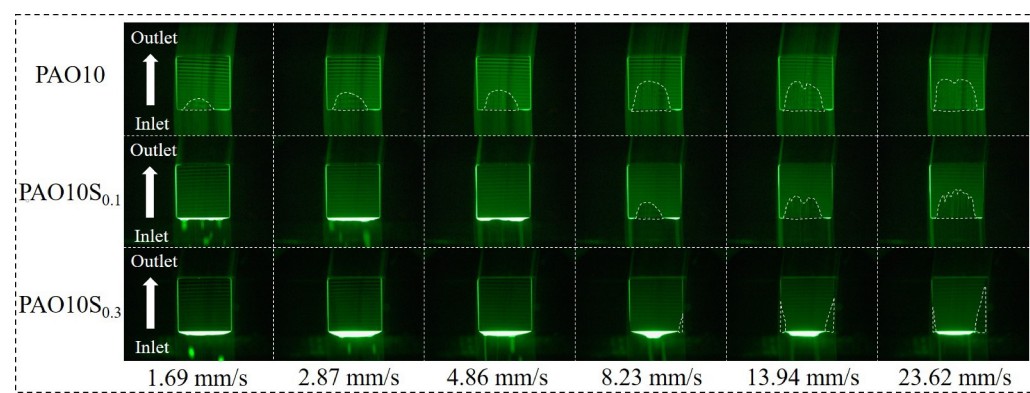

**Figure 5.** Fluorescence images of the oil reservoir at the entrance of the slider with speed, $w = 2$ N, $q = 2$ µL. (Note that the region of starvation is highlighted by dotted lines for clarify.).

The florescence images of PAO10 clearly show at the slider outlet that the lubricating oil was displaced to both sides of the lubrication track by the slider, forming the classical oil distribution feature of 'thin central oil layer bordered by oil ridges on the two sides'. Since the central oil layer of the lubrication track is thin, a small oil starvation area at the slider inlet readily occurred, even at the low speed of 1.69 mm/s; this is attributed to insufficient oil supply. The oil starvation area increased with the sliding speed.

However, PAO10$S_{0.1}$ and PAO10$S_{0.3}$ presented discrete stripe- or droplet-like oil supply at low speeds (as shown in the images of speed 1.69 mm/s and 2.87 mm/s), and there was accumulation of lubricating oil at the slider entrance. The oil starvation is

relatively small compared with the pure PAO10. The oil starvation started to appear at higher speeds with the increase in the mass fraction of stearic acid. For PAO10S$_{0.1}$, the starvation started to appear in the image of 8.23 mm/s. Oil accumulation at the entrance of the slider was obvious at low speeds (up to 4.86 mm/s). However, the time intervals for the successive slide-over may not be long enough to allow the oil bead up (dewetting) completely at high sliding speeds, such that oil beads or stripes are hardly seen in the images of speeds beyond 4.86 mm/s. At the speed of 8.23 mm/s, the oil reservoir at the slider entrance broke down, leading to the occurrence of an oil starvation area. For PAO10S$_{0.3}$, there was consistently a certain amount of lubricating oil accumulated at the slider entrance for the whole specified speed range. At the speed of 8.23 mm/s, it shows that more oil accumulated at the center than the two sides. Overall, under LLS conditions, the discrete oil supply induced by the adsorption of stearic acid greatly improves the conformal contact lubrication, as shown by the delay of the starvation occurrence to higher speeds.

In the slider-on-disc contact lubrication system with LLS, the oil displaced by the slider to the two sides forming oil ridges became the oil source from the return flow to the central track, i.e., replenishment. Figure 6 shows the distribution of oil reservoir at the side edge of the slider. PAO10 had a strong affinity, i.e., good wettability, with the glass surface. Thus, the oil reservoir at the sides of the slider, as shown in Figure 6a, is wide when comparing with the other two oil specimens containing a small amount of stearic acid (Figure 6b,c), as a result of the combined action of the side leakage and the displacement by the slider. Therefore, the lubrication track had a wider width than the slider. For the specimen oils containing stearic acid (PAO10S$_{0.1}$ and PAO10S$_{0.3}$), the oil spreading was diminished due to the dewetting phenomenon on the bearing track as shown in Figure 4b,c. The width of the oil reservoir at the side edges of the slider was significantly reduced (as shown in Figure 6b,c). Hence, only a small amount of lubricating oil can be observed on the slider side edges, and most of the lubricating oil remained in the track for lubrication.

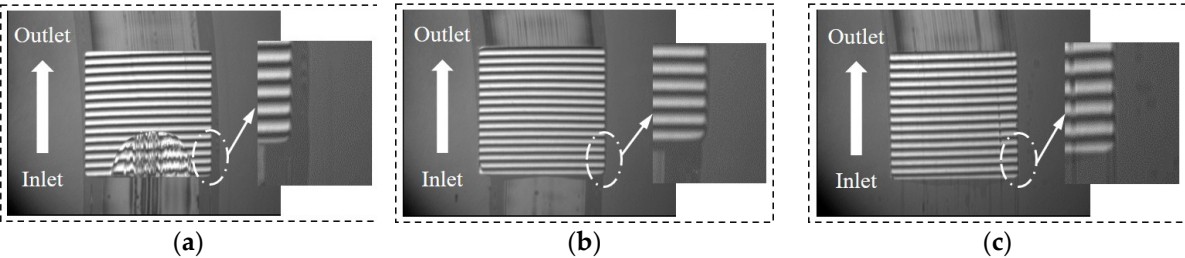

**Figure 6.** The distribution of the oil on the slider side edge, $w$ = 2 N, $q$ = 2 μL, $u_{\mathrm{d}}$ = 4.86 mm/s. (**a**) PAO10, (**b**) PAO10S$_{0.1}$ and (**c**) PAO10$_{0.3}$.

### 3.2. Effect of Stearic Acid Adsorption on Lubricating Oil Film

Figure 7 shows the interferograms of the lubricating oil film in the contact area. For PAO10, with the increase in sliding speed, the time allowed for oil replenishment from the side ridges became shorter, and the insufficient oil supply led to the continuous enlargement of the oil starvation area. At the same speed, the oil starvation area formed by lubricating oil with the stearic acid additives was smaller than that of the pure PAO10. Overall speaking, the degree of oil starvation is greatly reduced with the addition of stearic acid into the base oil.

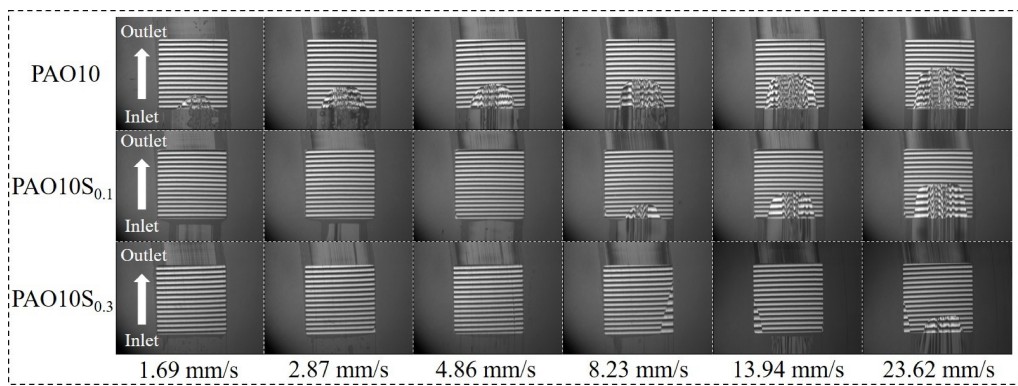

**Figure 7.** Interference diagram of the lubricant film in the contact area of the slider, $w = 2$ N, $q = 2$ μL.

Figure 8 depicts the variation of the minimum film thickness of PAO10, PAO10S$_{0.1}$ and PAO10S$_{0.3}$ with speeds for the oil supply of 2 μL and load of 2 N. The minimum film thickness established by the three lubricants increases with the sliding speed. At low speeds, the time allowed for the oil replenishment from the two side oil ridges is long and the size of the converged bearing gap, i.e., the required oil volume, is small. Therefore, the minimum film thickness achieved by the three different specimen oils had no great difference. However, the oil samples containing stearic acid produced higher film thickness than the pure PAO10 oil at high speeds. The film thickness increases with the loading of stearic acid in the lubricating oil. The stearic acid adsorption changes the surface property of the glass disc. The affinity between the oil and the adsorbed layer is weak, which thus promotes dewetting. Accordingly, the oil distribution on the lubrication track is characterized by discrete oil stripes (Figure 4b) or oil beads (Figure 4c) instead of a continuously oil film bordered by side oil ridges (Figure 4a). On one hand, the dewetting phenomenon attributed to stearic acid adsorption resulted in reducing the spreading or leaking of the lubricating oil to the two sides of the track. On the other hand, the discrete oil stripe- or bead-type supplied oil layer was also beneficial to the LLS lubrication, as schematically illustrated in Figure 9. The discrete oil supply mode made the lubricating oil participate in lubrication earlier at the inlet of the slider when compared with the continuous oil supplied layer for LLS lubrication. It generated a higher hydrodynamic effect and increased the thickness of the lubricating oil film.

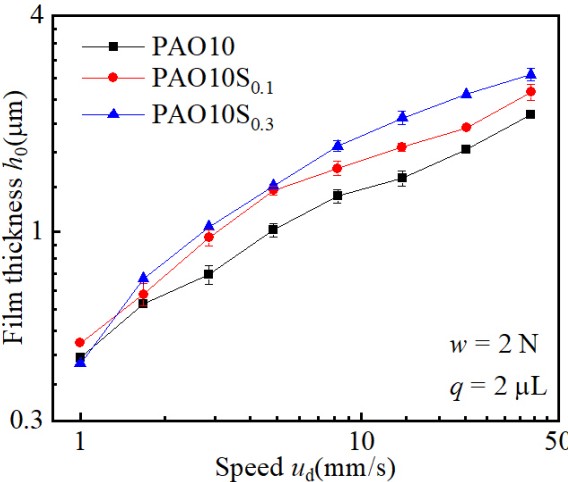

**Figure 8.** The curve of minimum oil film thickness with speed.

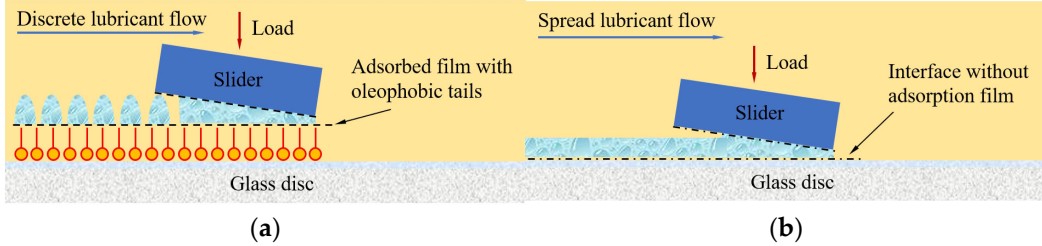

**Figure 9.** Lubricating oil flow model. (**a**) discrete state; (**b**) spread state.

Figure 10 shows the curves of the minimum film thickness formed by PAO10 and PAO10S$_{0.1}$ lubricating oils with sliding speed for the oil supply of 2 μL and the load of 4 N and 6 N, respectively. The film thickness of the lubricating oil containing stearic acid additives was higher than that of pure PAO10. For the load of 4 N (Figure 10a), the largest increase in the film thickness with the addition of stearic acid was 32.1% at 4.86 mm/s. For the load of 6 N (Figure 10b), the addition of stearic acid led to the largest increase in film thickness by 27.8% at 40 mm/s. These show that the discrete oil supply mode induced by stearic acid adsorption can also play a positive role in the oil film formation under LLS conditions.

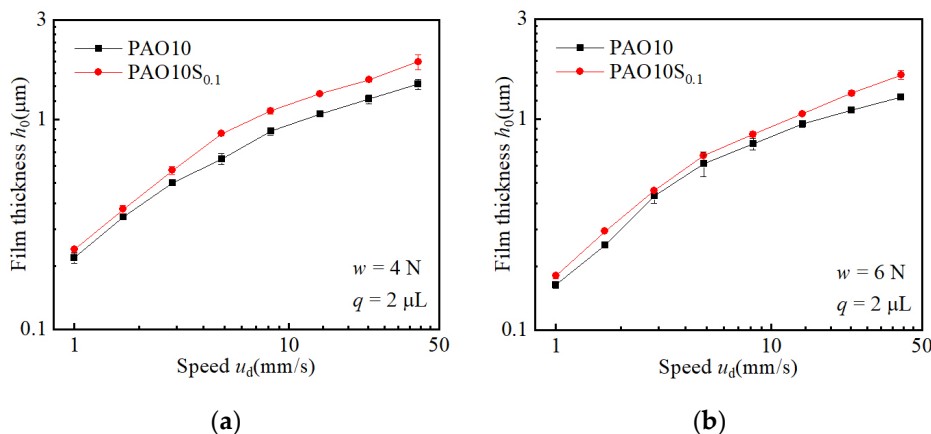

(**a**)    (**b**)

**Figure 10.** Variations of minimum oil film thickness with speed. (**a**) $w$ = 4 N; (**b**) $w$ = 6 N.

The comparisons of the lubricating film thickness under LLS (2 μL) and fully flooded (full) conditions for PAO10 and PAO10S$_{0.1}$ are, respectively, shown in Figure 11a,b. Figure 11a shows that, the PAO10 film thicknesses at LLS and full conditions were about the same at low speeds. The lubricating oil had sufficient time to reflow due to the low running speed. Accordingly, sufficient oil from the supply of 2 μL was entrained into the bearing contact to generate an oil film of the same thickness as the one with the fully flooded condition. The time allowed for the replenishment of PAO10 from both side ridges reduced with the increase in sliding speed, such that the degree of oil starvation was increased. The influence of oil supply on the oil film thickness begins to be obvious. The oil film thickness established by 2 μL lubricating oil was significantly lower than that of the full oil supply. However, for the lubricant of PAO10S$_{0.1}$ as shown in Figure 11b, an 'abnormal' phenomenon was observed at low speeds. For the oil supply of 2 μL, the oil film thickness was apparently higher than that of the full oil supply up to a certain speed. Figure 5 shows that PAO10S$_{0.1}$ accumulated at the slider entrance at low speeds because of the discrete oil supply layer due to dewetting. It is probably the reason for PAO10S$_{0.1}$ of 2 μL presenting higher oil film thickness than that of the full oil supply. Such oil accumulation never appears in any LLS test of the pure PAO10 oil. That is why the 'abnormal' higher film thickness with LLS is not seen in Figure 11a.

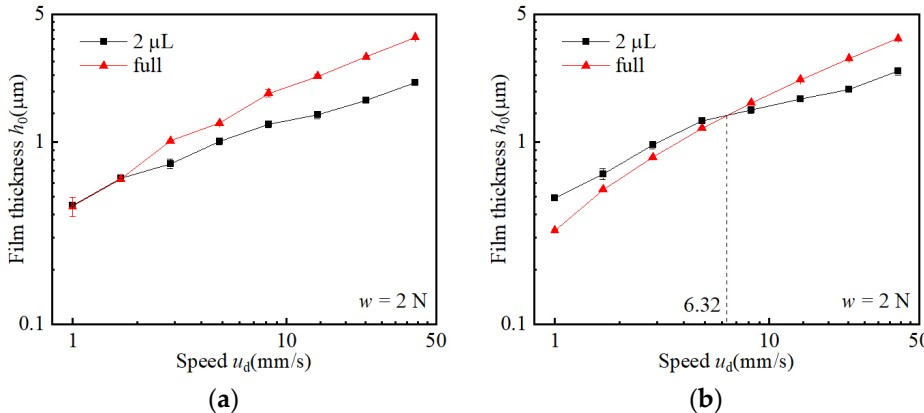

**Figure 11.** Variations of minimum oil film thickness with speed under different oil supply. (**a**) PAO10; (**b**) PAO10S$_{0.1}$.

From the image of PAO10S$_{0.1}$ at the sliding speed of 2.87 mm/s in Figure 5, the lateral profile of the oil accumulated at the slider entrance (i.e., the oil reservoir) was traced using the fluorescence method and plotted in Figure 12. Details about the application of fluorescence for thick oil film thickness measurements can be found in [26]. The oil supply was 2 μL. The lateral profile resembles approximately a quadrant of radius *r*. According to the general form of the Young–Laplace equation [27], the pressure difference between the inside and the outside of a curved boundary between two fluid regions is related to the shape of the boundary. If the surface of the oil reservoir at the inlet entrance is approximated as a cylindrical surface, the additional pressure at the inlet of the slider is:

$$\Delta p = p^{\alpha} - p^{\beta} = \frac{\gamma}{r} \tag{2}$$

where:

$p^{\alpha}$ is the pressure of the accumulated oil (Pa);
$p^{\beta}$ is the ambient pressure (Pa);
$r$ is the radius of the curved surface (m).

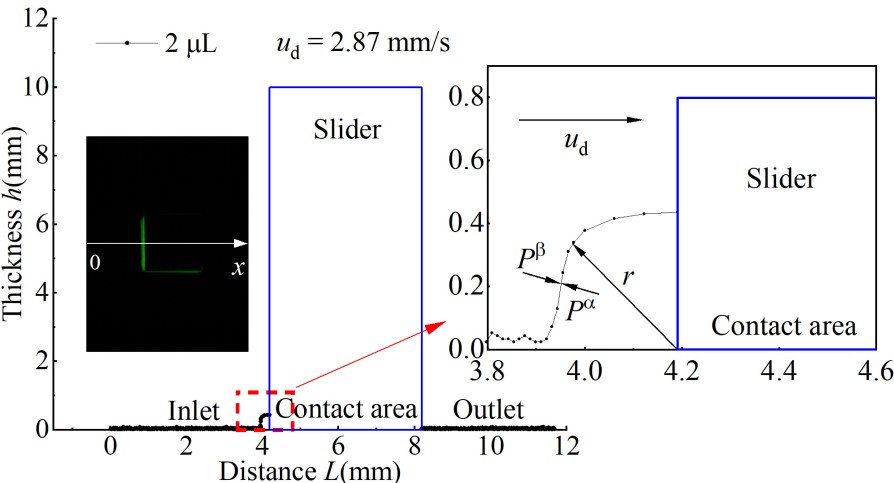

**Figure 12.** Accumulation of lubricant at the inlet of slider, *w* = 2 N.

When under the full supply condition, the supply oil is a continuous layer. Its free surface acquires theoretically an infinite radius of curvature. Equation (2) gives zero additional pressure. Thus, the pressure at the inlet of the slider is zero. However, when the oil supply is 2 μL, an oil reservoir of PAO10S$_{0.1}$ oil is formed, which produces additional

pressure under the action of surface tension. Thus, the pressure boundary condition at the inlet of the slider is changed. The additional pressure plays a positive role in the establishment of the oil film. Therefore, the 'abnormal' phenomenon of higher LLS film thickness appears as shown in Figure 11b.

To verify the hypothesis, the inlet oil reservoir is modeled as shown in Figure 13 and further theoretical analysis of hydrodynamic lubrication was carried out. If the pressure leakage at both sides of the slider is not considered [28], the Reynolds equation of incompressible steady-state slider lubrication is expressed as:

$$\frac{\mathrm{d}}{\mathrm{d}x}\left(h^3 \frac{\mathrm{d}p}{\mathrm{d}x}\right) = -6\eta u_\mathrm{d} \frac{\mathrm{d}h}{\mathrm{d}x} \tag{3}$$

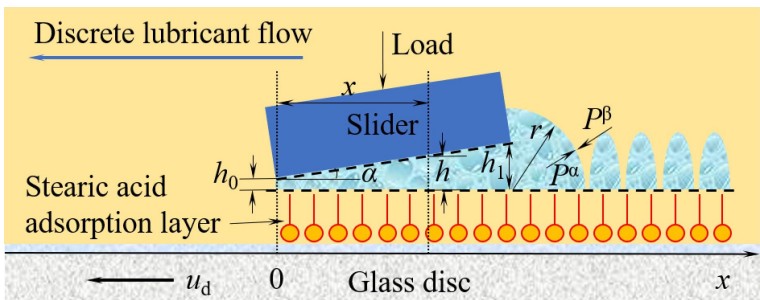

**Figure 13.** Accumulation mode of lubricant at the inlet of slider.

The boundary conditions of Equation (3) are: $h = h_0$ ($x = 0$), $p = 0$; $h = h_1$ ($x = B$), $p = \Delta p$. The bearing capacity per unit length of the slider $W$ is as:

$$W = \frac{6\eta u_\mathrm{d} B^2}{K^2 h_0^2}\left[\ln(K+1) - \frac{2K}{K+2} + \frac{\Delta p h_0^2 K^2(K+1)}{6\eta u_\mathrm{d} B(K+2)}\right] \tag{4}$$

where $K$ is the convergence ratio:

$$K = \frac{h_1 - h_0}{h_0} = \frac{h_1}{h_0} - 1 \tag{5}$$

When the oil supply is 2 μL and oil is accumulated at the inlet entrance, the pressure boundary condition at the slider inlet is finite, i.e., $\Delta p_{2\mu\mathrm{L}} > 0$. When the oil supply is sufficient such that a continuous oil supply layer occurs on the track, the Reynolds boundary condition is applied at the inlet, i.e., $\Delta p_\mathrm{full} = 0$. Equation (4) can be applied to both conditions. Considering the constant $W$ yields Equation (6). If the test conditions ($\eta$, $u_\mathrm{d}$, $B$) are constant and the same convergence ratio $K$ is considered, the minimum film thickness of 2 μL oil supply is higher than that of the full supply condition, $h_{0(2\mu\mathrm{L})} > h_{0(\mathrm{full})}$, since $\Delta p_{2\mu\mathrm{L}} > \Delta p_\mathrm{full}$.

$$\frac{6\eta u_\mathrm{d} B^2}{K^2 h_{0(2\mu\mathrm{L})}^2}\left[\ln(K+1) - \frac{2K}{K+2}\right] + \frac{\Delta p_{2\mu\mathrm{L}} B(K+1)}{K+2} = \frac{6\eta u_\mathrm{d} B^2}{K^2 h_{0(\mathrm{full})}^2}\left[\ln(K+1) - \frac{2K}{K+2}\right] + \frac{\Delta p_\mathrm{full} B(K+1)}{K+2} \tag{6}$$

In the slider-on-disc contact lubrication system under LLS conditions, a pressure difference between the inner and outer surfaces of the lubricating oil reservoir at the slider entrance exists due to surface tension. Thus, the pressure boundary condition at the slider inlet is changed from a null to a finite value; this is beneficial to the lubricating film formation. It is in agreement with the results obtained in the experiments.

Similar phenomena are also observed from the experiment with the lubricant PAO10S$_{0.3}$ as shown in Figure 14. The 'abnormal' phenomenon is that the LLS oil film is of higher thickness than the full oil film at low speeds. Figure 14 shows that the oil film thickness established by 2 μL PAO10S$_{0.3}$ was higher than that when using the full oil supply for

speeds less than 5.73 mm/s. The result is consistent with that of $PAO10S_{0.1}$. The 'abnormal' phenomenon is attributed to the effect of the surface tension of the oil reservoir that is formed by the accumulation of discrete oil droplets at the slider entrance. The distribution of $PAO10S_{0.3}$ on the bearing track at LLS conditions is also characterized by the discrete oil supply mode as shown in Figure 4c. This finding highlights that in a certain speed range, the lubricating film formation capability of LLS is higher than that under sufficient oil supply in the slider bearing contact due to the stearic acid adsorption.

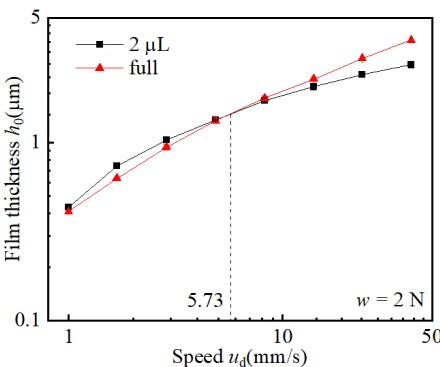

**Figure 14.** Variations of minimum oil film thickness with speed under a different oil supply, $PAO10S_{0.3}$.

### 3.3. Surface Adsorption Characterization of Stearic Acid Additives

The morphology of the stearic acid adsorption on glass and steel surfaces was probed by an atomic force microscope (AFM) (Bruker (Beijing) Technology Co., Ltd., Beijing, China). The experimental steps are shown in Figure 15. A glass block that was coated with the semi-reflective bi-layers (Cr and $SiO_2$, dimensions: 30 mm × 30 mm × 10 mm) as the glass disc used in the experiments and a steel slider (dimensions: 4 mm × 4 mm × 10 mm) were cleaned by plasma sputtering with oxygen for 2 min. They were then soaked in a sealed bath of PAO10 plus 0.1% stearic acid for 6 h at the ambient temperature of 22 °C to allow full adsorption of stearic acid on the surfaces of the specimens. The samples were rinsed with n-heptane afterwards and then dried with nitrogen. The morphology of the sample surfaces was obtained using AFM (Tapping Mode).

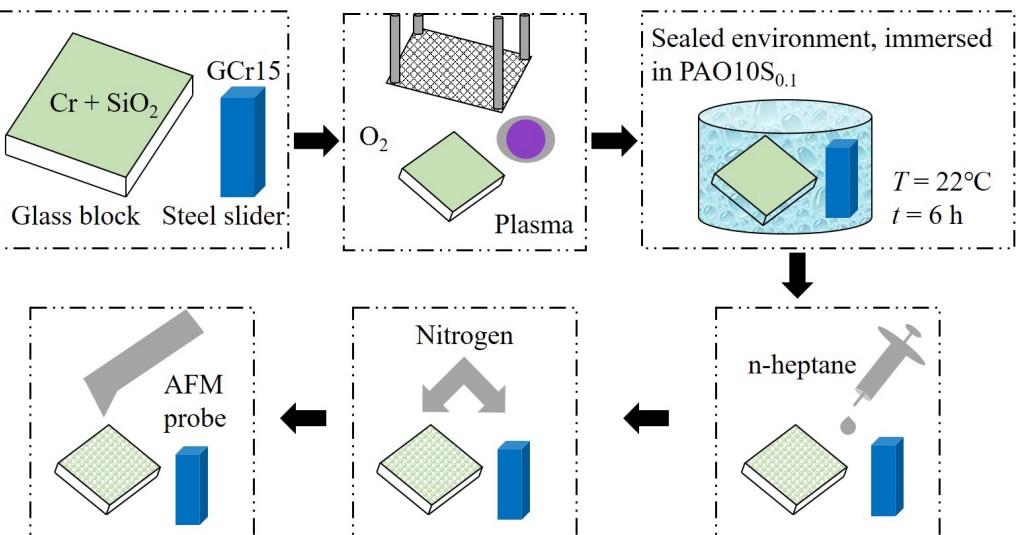

**Figure 15.** Sample preparation and detection process.

The comparison of the sample surfaces before and after the adsorption treatment is shown in Figure 16. The surfaces of the untreated glass block (Figure 16a) and steel

slider (Figure 16c) are relatively smooth, whereas the treated sample surfaces look rough (Figure 16b,d). The morphology of the treated glass surface (Figure 16b) is characterized with nano-sized granular bulges, while that of the treated steel surface presents a brush-like structure (Figure 16d). Figure 16 indicates that both the glass and steel surfaces were coated with adsorption layers after the surface treatment.

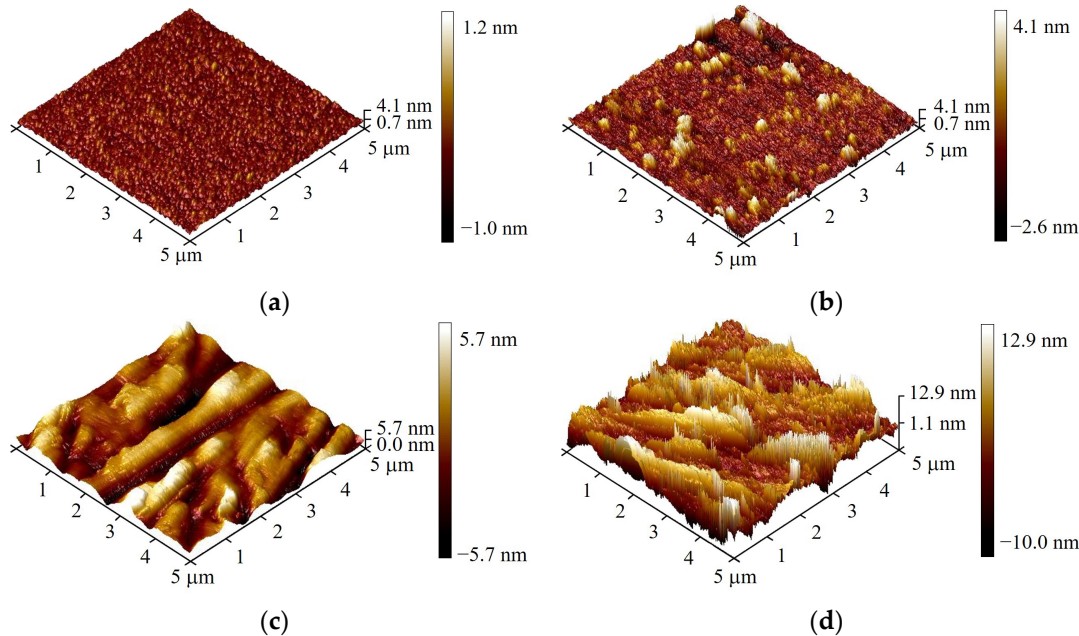

**Figure 16.** AFM surface morphology measurement results. (**a**) original glass block; (**b**) adsorbed glass block; (**c**) original steel slider; (**d**) adsorbed steel slider.

The stearic acid adsorption layer on the surface of steel slider was characterized using a Fourier transform infrared spectrometer. Steel sliders of the same material and size were cleaned thoroughly and then respectively immersed in PAO10, PAO10S$_{0.1}$ and PAO10S$_{0.3}$ lubricating oils for 6 h. They were rinsed with n-heptane and dried with nitrogen. Finally, the surface was detected. The results of the Fourier transform infrared spectrometer are shown in Figure 17. For the steel surface soaked in PAO10S$_{0.1}$ and PAO10S$_{0.3}$ lubricating oils, the stretching vibration of characteristic infrared absorption peaks corresponding to C=O was observed at 1648 cm$^{-1}$ and 1457 cm$^{-1}$, indicating the existence of –COOH functional group [29]. Thus, the adsorption of stearic acid molecules on the steel surface is validated.

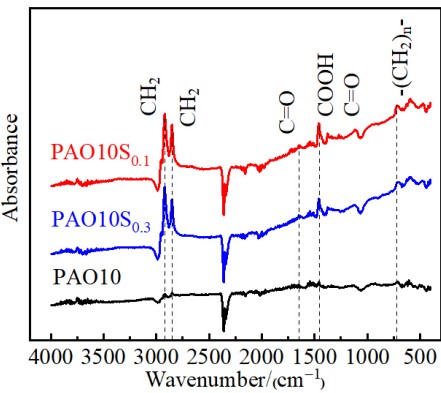

**Figure 17.** Fourier infrared spectroscopy analysis of steel slider surface.

## 4. Conclusions

In this paper, the effect of stearic acid adsorption on the morphology of the distributed lubricating oil on the lubrication track and the oil film thickness were investigated using an optical slider-bearing test rig and fluorescence measurement apparatus. The conclusions are as follows:

1.  Under LLS, with the addition of a small percentage of stearic acid to the base PAO oil, the lubricating oil presents a discrete stripe or droplet distribution owing to the stearic acid adsorption layer that changes the wettability of the surface.
2.  The discrete oil supply mode induced by stearic acid adsorption enriches the LLS lubrication, via the early contacts between the entrained oil and the slider surface.
3.  The accumulation of lubricating oil at the slider entrance caused by the discrete oil supply mode changes the pressure boundary condition at the bearing inlet, which plays a positive role in promoting the film formation capability.
4.  LLS lubrication can be improved by the adsorption of stearic acid, via the dewetting phenomenon. This new mechanism is worth further exploration to identify appropriate additives for the enhancement of LLS lubrication.

**Author Contributions:** Conceptualization, F.G., P.L.W. and Y.J.; methodology, Y.J. and F.G.; validation, Y.J.; formal analysis, Y.J. and Z.J.; investigation, Y.J., F.G. and P.L.W.; resources, Y.J.; data curation, Y.J. and Z.J.; writing—original draft preparation, Y.J.; writing—review and editing, F.G. and P.L.W.; visualization, F.G. and P.L.W.; supervision, F.G. and X.L.; project administration, F.G.; funding acquisition, F.G. All authors have read and agreed to the published version of the manuscript.

**Funding:** This research was funded by the National Natural Science Foundation of China, grant number 52175173, and the Research Grants Council of Hong Kong, project number CityU11216619.

**Data Availability Statement:** Not applicable.

**Conflicts of Interest:** The authors declare no conflict of interest.

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
