# Peer review of "Experimental Study on the Lubrication Enhancement of Slider-on-Disc Contact by Stearic Acid Adsorption under Limited Lubricant Supply"

_lubricants, doi:10.3390/lubricants10120353_

Round 1
Reviewer 1 Report
The paper presents an interesting approach based on the Experimental Study on the Lubrication Enhancement of Slider-on-Disc Contact by Stearic Acid Adsorption under Limited Lubricant Supply. However, the innovation of the current research work should be further highlighted and emphasized. At the same time, the authors should consider the following comments to greatly improve the quality of the paper.
1. In the abstract, add a final statement that highlights the importance of this research and its possible potentials. Also, introduce the problem in the initial lines of the abstract.
2. The introduction needs to be improved by relating to the mechanics of the studied materials and their mechanical characteristics. The references to be included are: 10.1177/00219983221141154, 10.1177/07316844211051733, 10.1016/j.polymertesting.2017.09.009, 10.1016/j.compstruct.2021.114698
3. Kindly add a table that describes the main physical and chemical properties of the raw materials used in this study.
4. Were the preparation methods described by the authors come in accordance with a certain standard or do they follow previous procedures?
5. For the sliding tests, what was the reason for the specified test conditions in this research? Do they relate to a specific application?
6. How many samples were used per configuration for the sliding test per testing condition?
7. The conclusion needs to be modified to summarize the research outcomes in short statements with clear observations.
Reviewer 2 Report
The submitted research article is well written. The content of this paper has high significance. The presentation quality is very good. I recommend to accept this paper after the following minor changes.
1.Why the authors have conducted the sliding experiments under the application of low load. This must be clearly mentioned in the manuscript.
2. Whether the Figure-1 (line 137) and Figure-12(line 337) collected from reference 24 and 26 ? The figure-12 is your experimental findings. Revise this, if am not wrong.
3. It is fine that authors have shown the AFM images of the adsobed glass block and steel slider. Along with this it will be good if the authors can show the SEM/ Optical microscope images of the glass block and the steel slider before and after the adsoption.
4. A broad explanation is neccessary to justify how the adsorbed layer promotes dewetting.
Reviewer 3 Report
Please read the comments

Reviewer 4 Report
In this study, the effect of stearic acid adsorption on the morphology of the distributed lubricating oil on the lubrication track and the oil film thickness were investigated by the optical slider-bearing test rig and fluorescence measurement apparatus. It was found that the stearic acid adsorption layer weakens the wettability of the lubrication track and induces the ‘dewetting’ phenomenon of the lubricating oil. The authors adopt the dewetting effect and the generated discrete distribution in the form of stripe or droplet can improve the tribological performance under starvation lubrication. It is an impressive work and inspires me a lot. This manuscript can be accepted for publication as it is.
